# Ageist No More: Interprofessional Training for Undergraduate Healthcare Students

**DOI:** 10.3390/geriatrics7010017

**Published:** 2022-02-07

**Authors:** Aniela Mendez, Mildred Lopez, Karina Rodriguez-Quintanilla, Belinda Carrion

**Affiliations:** Escuela de Medicina y Ciencias de la Salud, Tecnologico de Monterrey, Monterrey 64710, NL, Mexico; aniela.mendez@tec.mx (A.M.); mildredlopez@tec.mx (M.L.); karina.rodriguezq@tec.mx (K.R.-Q.)

**Keywords:** prejudices, health education, intergenerational interventions, ageism, older adults, educational innovation

## Abstract

Ageism seeps deep into our society, whether in law, policies, or healthcare practices it segregates individuals based on their age. The aim of this work was to evaluate the impact of an educational strategy in ageist attitudes against older adults in healthcare undergraduate students. A five-week intervention: Healthy environments and self-care for the older adults was implemented. To assess the impact of this strategy in ageist attitudes in participants, a simulated consultation with an older adult was conducted. Participants’ perspectives on the experience were collected using an online survey. One hundred and thirty-eight undergraduate students from health programs were included. They highlighted growth in the understanding of the normal aging process and the prejudices that surround aging. During the role-play activity, participants identified communication, empathy, and professionalism as the abilities developed with this strategy and the need to show empathy and avoid prejudice against older adults in their clinical interactions. Educational interventions are a great tool to promote cultural changes, diminish prejudices and misconceptions of ageism in future healthcare professionals.

## 1. Introduction

Quality of care has improved in the past decades, as a consequence, the life expectancy of individuals has risen considerably [1], this demographic change comes with challenges to countries and brings the need to change their economic and social politics to promote inclusiveness to older populations, it includes healthcare and other human rights. However, the prejudices surrounding aging and the discriminatory attitudes that exist towards older adults persist and need to be ceased to provide equitable access to health. According to the World Health Organization (WHO), ageism describes the spectrum of attitudes that encompass, how we think, how we feel, and how we act towards ourselves or others depending on age [2]. These misconceptions influence how we relate to others, imagine not hiring a candidate because of their youthful appearance, or being pushed out of a job because of being ‘too’ old. These everyday battles are the ones we must fight to unroot attitudes that are deeply weaved in how systems are organized.

Discriminatory attitudes towards a person based on their age and negative perceptions about aging seep deep into our society, the worst part might be the individual not being aware of these attitudes. Negative attitudes and stereotypes are often fixed generalized beliefs with no rational explanation, so, to be aware of irrational fixed behavior, first one should identify and define this unconscious idea [3]. Negative attitudes based on elder stereotypes assume lower ability or being incapable of independence [3]. Previous studies show that when exposed to older adults, young people switch their speech to patronizing and stereotypical [4]. Others believe that older adults have lost their will to improve their quality of life or desire for leisure and recreational activities [5]. These beliefs often drive the health professional to impose decisions on the elder without considering their opinion. Negative and discriminatory attitudes in ageism tend to bias clinical decisions not considering the patient’s autonomy and imposing management without patients’ compromise [5].

Ageist attitudes in healthcare workers come with particular and considerable risks. Older adults may not have proper access to healthcare, for example, in nursing, several authors report a reluctance to work or care for older adults as there is a perception that this career path represents lower professional prestige [6]. In the case of medical residents, a large study found that they show mostly negative perceptions and attitudes towards aging [7]. The prejudices surrounding aging cause a barrier to effective relationships with older adults [8], thus affecting the doctor–patient relationship.

### 1.1. The Consequences of Ageism

Whether we notice or not, ageism seeps deep into our society, it can be seen in laws, marketing, policies, or healthcare practices that segregate individuals based on their age. These negative perceptions surrounding the process of aging downsize the person and affect them in multiple dimensions. The World Health Organization (WHO) defines health as the state of complete physical, mental, and social well-being and not merely the absence of disease [9]. The effects of ageism are vast and can affect all the dimensions of health, the physical, social, and mental well-being.

In the physical sphere of well-being, persons that have ageist beliefs are less likely to take care of their health or maintain healthy living habits that reduce the risk of illness [10]. Physicians may also show less patience or optimism when treating older adults and exhibit patronizing attitudes when attending older adults [11], they may also fail to include older patients in the decision-making process, a crucial part of the desired modern healthcare systems.

In the social sphere, ageist attitudes in healthcare can limit treatment and information access [12] and put patients at greater mortality risk [10]. In 2020, countries like the USA reported an increase in annual costs in healthcare up to 63 billion USD [1]. Decreased support to fulfill their potential better standard living [13].

In the mental sphere, ageist attitudes to older adults have deep psychological consequences [14], these attitudes have been linked to depression, poor mental health outcomes [15], and lower overall wellbeing [16]. Older adults may also have limited access to certain psychological treatment amid misconceptions about their later potential in life [17]. Figure 1 depicts the impact of negative attitudes in the different spheres of health, and even though separately grouped, it is important to establish the interdependence and repercussions of the negative effects with the other domains.

This reality has drawn attention to international associations such as the United Nations (UN), WHO, and the United Nations Department of Economic and Social Affairs (UN DESA) who have called for strategic actions to raise awareness and combat discriminatory attitudes [2].

### 1.2. Strategies to Combat Ageism

To address the consequences of ageism worldwide, the WHO has proposed three main strategies: educational activities, intergenerational interventions, and focusing on policy and law [2]. Several authors have found that one of the main causes of suboptimal care for older patients is related to the lack of training of health personnel [18]. Other reports have highlighted the importance of designing and implementing strategies that rely on interventions that educate about the normal process of aging, and intergenerational approaches that facilitate contact with older adults [19]. Intergenerational strategies have been pointed out as the best to promote positive interactions with older adults and significant learning experiences [14].

Patient-centered learning is one of our most important strategies that center on the elder as the reason to mitigate the negative against them [6]. According to Koren et al. 2008 [20], promoting positive attitudes early in the training of undergraduate medical students is vital to counteract negative attitudes towards older adults. Unfortunately, these occur too late in the curricula [21]. While many studies urge the need to design and implement strategies that address the consequences of ageist attitudes towards older adults, few take focus on creating awareness for clinical practitioners and undergraduate health students on this matter [19].

Patient-centered clinical care underlies certain skills to convey a nourishing rapport between health professionals and patients. One of them is communication skills that hold positive repercussions especially in patient satisfaction and adherence to treatment [22]. It is also important to build a trustful relationship with patients [23]. In order to construct a solid doctor–patient relationship, communication skills interact with empathy since it allows one to dive into patients’ expectations, fears, or feelings, favoring healthy relationships [24].

Patient-centered models also require professionalism skills. This is related to professional behavior based on ethical conduct and responsibility towards the welfare of the patient [25]. Patients who experience professional interactions are more likely to explain symptoms, provide relevant information, and adhere to treatment [26].

### 1.3. Our Educational Strategy

A pilot implementation strategy to battle ageism was created in the Healthy Environments and Self-care for the Older Adults module. This strategy was implemented in the School of Medicine and Health Sciences of Tecnologico de Monterrey, an institution that has adopted Challenge Based Learning as an educational model [27]. This five-week module is taken by third-semester students from health programs (psychology, medicine, nutrition, biosciences, and odontology). It includes two main parts: the challenge, and the theoretical and practical modules. At the end of the five-week period, students are enrolled in a role-play simulation. Figure 2 depicts a general outline of the five-week strategy.

As this strategy took place in Mexico, it was conducted in Spanish, the native language of the students and teachers. The temporality of the experience was during the SARS-CoV-2 pandemic; hence it was conducted online using different video conferencing tools, and a Learning Management System (LMS) to organize content and for students to interact asynchronously.

### 1.4. The Challenge

The challenge represents a real-world problem where students are asked to design and propose solutions for improving the quality of life for older adults from a low-income marginalized community with identified economical, educational, and social needs. Students were grouped in teams (5–7 members) and worked, guided by a teacher, to conceive, develop and pitch the solution. The students have a Challenge advisor, who provides guidance and feedback throughout the five-week period.

The sequence structure of the challenge is as follows, first, students choose a domain of the quality of life of older people where they want to focus. This is undertaken with the help of the quality-of-life assessment questionnaire to older adults, widely recognized in the field as WHOQOL-OLD [28]. This instrument was designed using a Likert scale with six different areas: sensory abilities, autonomy, past, present, and future activities, social present and future activities, social participation, death, and intimacy. During the five weeks, students work together to create the strategy, stages of implementation, the material and human resources required, and the indicators to measure the impact of their proposal on the quality of life of older adults. Students must present the solution after the five-week period to evaluators who give feedback for the feasibility and adequacy of the projects.

### 1.5. The Theoretical and Practical Modules

The theoretical and practical modules are designed to nurture and provide a scientific basis to the student’s proposed solutions to the challenge with different activities. These are conducted along with the challenges in two-hour periods. They are focused to cover the most important topics about the aging process, healthy aging, comprehensive care to older adults, integrated care to older people [29], and psychological and quality of care of older adults. Table 1 shows with greater detail the design of the five-week module, including the challenge, practical and theoretical modules.

## 2. Materials and Methods

Seeking to assess the effectiveness of this educational strategy on ageist attitudes, all students that completed the module were enrolled in a simulated scenario. Even though this activity was not considered mandatory for the course, students were encouraged to take part in this dynamic as a formative activity. None refused to participate as they declared they were eager to receive a more practical experience despite being in a remote learning format.

The sample consisted of 138 undergraduate students from the psychology, medicine, nutrition, bioscientist, and odontology programs. The participants were enrolled in the third semester, and ages varied from 19 to 20 years, gender was not investigated.

### Description of the Role-Play

Leveraging the use of the videoconferencing tool to conduct the implementation, students were assigned to Breakout Rooms (BOR) for 40 min in three roles: health professional, patient, and observer. Scripts were given prior to the class and according to their role. The case in the simulated scenario was a primary care consultation with a retired older adult with insomnia and depressive symptoms, the complaints were exacerbated after retirement. Figure 3 depicts a visual overview of the simulated scenario activity including the delivery of the script and the online survey completed afterward.

*(a)* 
*The health professional role*


The student in this role was instructed to interrogate the patient with focusing on three dimensions: the social, the physical, and the psychological spheres. In the social sphere, the student in the health professional role had to investigate recreational activity, in the physical sphere they focused on investigating sleep habits, caffeine intake, and physical activity. Lastly, in the psychological sphere they investigate mood changes, lack of interest in daily activities, students were also given the option to implement the Patient Health Questionnaire (PHQ-9) [30] to deepen their understanding of the patient’s mood. After the consultation, they must explain to the patient that the symptoms presented were caused by the aging process and give a few recommendations to improve life quality.

*(b)* 
*The patient role*


The student in the patient’s role was instructed to complain to the attending health professional about insomnia. The script explicitly stated that the patient had encountered previous physicians who attributed his aches to his aging process, causing emotional discomfort and confusion to the patient. The *patient* had to express that previous health professionals gave him the idea that aging is a negative process, and to show discomfort due to the feeling of not being taken seriously. He attended the actual consultation seeking a diagnosis, treatment, and support.

*(c)* 
*The observer role*


The observer had a more passive role in the simulated consultation. Their scripts had the details of the other roles, and an instruction to observe and take notes on the interaction. The observer had a rubric formulated as a five-point Likert scale, to assess key elements of a successful consultation, taking focus on basic communication aspects. The rubric included questions like:Did the *health professional* show interest in the *patient’s* feelings?Did the *health professional* include the *patient* in therapeutic plans?Did the *health professional* use clear and comprehensive language?Did the gestures match the tone of the voice?Did the *health professional* ensure a comfortable atmosphere for the *patient*?

Even though this activity was not part of the student’s grades, the rationale for having observers in the team was to include an impartial role in the simulated scenario and assess if our role-play was promoting the right abilities.

One question focused on the learning experience, they were given a list of nine competencies and they had to identify up to three that this educational strategy helped them develop the most. The competencies included were: critical thinking, professionalism and ethics, teamwork, empathy, communication skills, resilience, information management, ability to control stress, and clinical skills. These competencies have been declared as core competencies for the medical student by many international learning frameworks [31].

The survey also included an open-ended question: *What did you learn from this experience?* This question included seeking their thoughts and their insights on the impact of the experience in their training and reflecting on their attitudes towards older adults. A content analysis strategy (qualitative approach) was used to understand participants’ responses and categorize them into codes. These categories were also analyzed and interpreted regarding the frequency in which they were mentioned. Three researchers independently read the responses and assigned codes, later reaching consent on descriptors and categories.

## 3. Results

Our sample consisted of 138 undergraduate students from health careers, all the students responded to the survey. After the 40 min of simulated consultation, the participants completed an online survey inquiring about their experience with the role-play.

First, participants were asked to select up to three competencies that the educational strategy helped them develop the most. It is important to note that some participants highlighted one, others two, and other three competencies. From the options provided, the participants identified communication which received 69 mentions, empathy with a frequency of 63, and professionalism with 56 mentions as the most relevant. Results are presented in Figure 4.

The distribution of the participants was as follows: 11.4% as health professionals, 33.3% as observers, and 55.3% as patients. Their responses have been encouraging as they valued the educational strategy as a significant learning experience. According to the results, the role that the participants portrayed influenced the impressions they had of the impact of the experience. As part of the analysis, researchers identified several codes: clinical needs, doctor–patient communication, prejudice against older adults, empathy to older adults, and validating emotions. Brief descriptions of the codes are presented as follows:Clinical needs referred to descriptions about the simulated scenario contribution towards the development of practical skills and the actions that a clinician can include to show dignity in patient-centered care.Doctor–patient communication refers to descriptions of the different critical moments in a clinical interview. It also referred to elements of communication with patients that can build trust in a respectful interaction in a clinical encounter.Prejudice against older adults referred to descriptions where participants observed potential ageist attitudes or actions towards the patient role or descriptions of previous experiences where their relatives experienced ageism firsthand.Empathy to older adults encompasses descriptions of students experiencing walking a mile in the patient’s shoes and the attention to demonstrating empathy through their discourse.Validating emotions includes taking into consideration the patient’s feelings when exposed to ageist attitudes. The participants in the patient role described the emotional validation they needed when exposing their different concerns, feeling vulnerable to the judgment of their pain to be classified as untrue.

As shown in Table 2, participants in the health professional role highlighted the impact on their clinical skills and doctor–patient communication. Nevertheless, participants either in the observer or patient role, describe similar highlights of the impact in empathy and avoid prejudice against older adults in their interactions.

Regarding participants’ reflections on their attitudes towards older adults, they highlighted growth in their understanding of ageism and the prejudices that surround aging, as the weeks in the educational experience passed, they would integrate an appropriate vocabulary to refer to the normal aging process and to avoid ageist vocabulary. The breaking point for many students came with the role-play dynamic of the consultation with the older adult. They pointed this activity as eye-opening as far as ageist attitudes in the health personnel regard. Especially for the students in the patient and observer role, who described clearly the activity helped to understand the importance of being empathic and that many behaviors to older adults are normalized and are prejudices that must stop, as you can see in Table 2.

An interesting element that was also mentioned by participants was their experiences of caring for older adults that are family members, some mentioned recent diagnosis of Alzheimer’s or accidents at home and felt unsure on how to care for them. They mentioned how the educational experience gave them resources to discuss openly with family members and describe feasible proposals to adapt home environments to them.

## 4. Discussion

As we prepare the health professionals for the future, we must align to social needs, and whether consciously or unconsciously, ageist attitudes still permeate our society. Our study sought to address the WHO’s urgent call to action on bringing down the stereotypes and misconceptions revolving around the aging process. Aligning to its recommendations, our work focused on strategies to educate health professionals to avoid ageism in their practice. Previous authors have highlighted the importance of designing educational interventions that promote a healthy concept of aging [19] and the importance of doing so early in the formative years of medical education [6].

During this five-week intervention, students were able to understand the physiology of aging, apply quality of life scales, and successfully identify possible areas of intervention to improve life quality for older adults.

As the results show, students highlighted the experience as it contributed to their training. Particularly communication, empathy, and professionalism, the top skills mentioned by the participants, are the key skills for health practitioners to positively impact the life quality of an older adult [14]. Effective communication has been repeatedly pointed out as one of the pillars of a healthy doctor–patient relationship [22,32,33], and in parallel to the curricular education, we aim to form compassionate and empathetic health care workers, capable of considering the emotional context of each one of their patients to ensure holistic care [34]. Professional attitudes in the clinical setting also reinforce the doctor–patient relationship, and altogether these abilities can be helpful especially in the care of older adults. Fostering these three abilities will enrich patient-centered education models.

Another interesting finding was the perception of the abilities developed depending on the role the students had. While students in the health professional role highlighted the practice of clinical abilities and the importance of doctor–patient communication, the results show that students in the patient and observer roles raised more awareness on ageist attitudes and generated more empathy towards older adults. Then, although students in the observer role may seem passive and we could question the utility of including them, their responses in the survey indicate otherwise. They benefit from the interaction by observing and by providing feedback to their peers. Their reflection of learning matches the ones of the students in the patient’s role. Future implementations will include students rotating through the different roles so they can gain a 360° perspective.

Later implementations need to design a training that has an incremental difficulty level. At the beginning of the training, participants might have several role-play simulations, then they can take part in examinations with standardized patients and later take part in student-run clinics with real patients. These real-life contexts could promote meaningful learning, and address the effects of these in the long term. Furthermore, the design of the cases and scripts can include more complex cases for students to solve.

Future studies could also be undertaken with students from different years of the health programs to explore the influence of their maturity as clinicians in their attitudes toward older adults. Another important variable to explore is if the virtual environment and the presence-based settings have comparable effects on students’ perceptions. One of the strengths of this work is that it is one of the most comprehensive educational interventions focused on aging and ageism that exists to undergraduate healthcare professionals, not only for medical careers but psychology, nutrition, odontology, and biomedicine to name a few. It helps to improve the aging perception of all healthcare professionals; this strategy has the potential to improve itself and to be replicated and adapted to others to make it more powerful.

## 5. Limitations

Even though this strategy had a good response between students, one of the main barriers for it to be part of the formal curriculum is that it requires active engagement by the teachers and the students. Furthermore, time could be a concern. This kind of activity requires exhaustive planning and effort from teachers and students; however, it is our opinion that the benefits that students gain—reported in this work—surpass the limitations. This strategy was designed in a challenge-based model which provides the optimum context for the module. This may pose challenges in the implementation of educational institutions with another educational model; nevertheless, this could be adapted to other educational environments.

The actual health crisis due to COVID-19 also limited this activity. Despite this, videoconference tools and collaborations from teachers and students made it possible. The face-to-face setting would be the optimal environment for these educational strategies, where students could benefit from interaction with each other and direct feedback from teachers. There are elements that could not be represented in the simulation, such as when the person enters the clinical office, the need to move the chairs or furniture to conduct a clinical examination, and the anxiety that the first clinical encounters might represent to our students.

Another limitation was the fact that Challenge advisors did not complete the survey to see what abilities they thought the students developed more during this simulated scenario. It would be interesting to contrast their appreciation with the students’ and see if they perceive the same learnings.

## 6. Conclusions

Our findings show that educational interventions are of great value in creating awareness and changing prejudices in ageist attitudes among healthcare students. This work represents an innovative educational strategy focused on improving the quality of life of older adults. By training and engaging undergraduate students early in their programs, universities can honor their commitment towards the community and have a social impact. As students gain an understanding of the biological, psychological, and social features of aging-particularly through peer role-play-they experience firsthand how their future patients might feel about receiving care. Whether they are preparing to be physicians, nutritionists, psychologists, or dentists, these interventions help to achieve health professionals with the tools to care for the older persons without stigma, prejudice, and discrimination, thereby reducing stereotype threat to older adults and then, offer truly patient-centered, interprofessional care.

Strategies, like the one discussed, have an impact in three domains: first giving the experience of a clinical encounter to undergraduate health students in a safe, controlled manner, and creating awareness among them of the abilities required to treat a real patient. Second, strengthening the core and relevant abilities to develop a healthy doctor–patient relationship like communication, empathy, and professionalism. This will add and improve their approach to an older patient. And third, by making students *walk a mile in the shoes* of the patient, or to experience-with a glimpse-the reality of how older patients may feel when exposed to certain behaviors will shape their future practices and make them better health professionals.

Society is the reflection of many factors, including, social, economic, and familiar contexts. If a true cultural change is needed, such as the one in aging and stereotypes that surround older adults, we must pursue changing the misconceptions that seep deep into our society. Education is our most powerful tool to change misconceptions about the aging process. This, in the long term, may be the key to ageist no more.

## Figures and Tables

**Figure 1 geriatrics-07-00017-f001:**
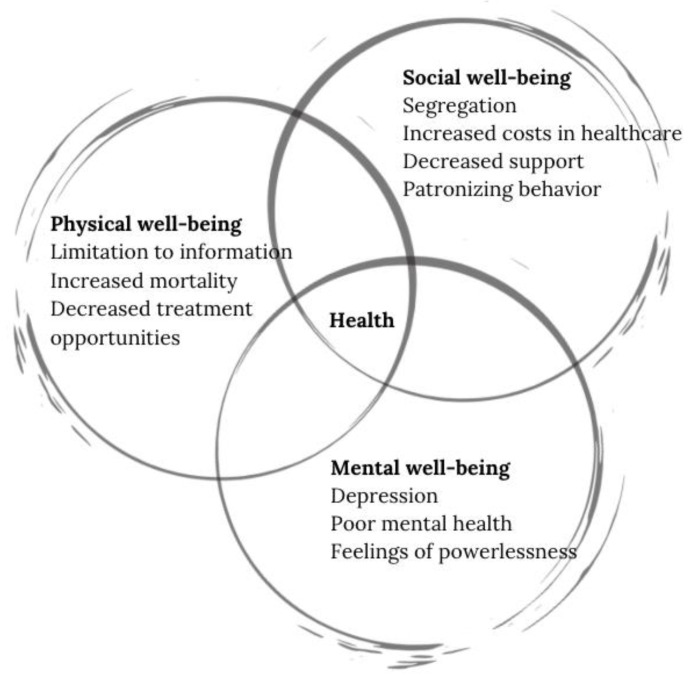
Impact of ageist attitudes in the different spheres of health.

**Figure 2 geriatrics-07-00017-f002:**
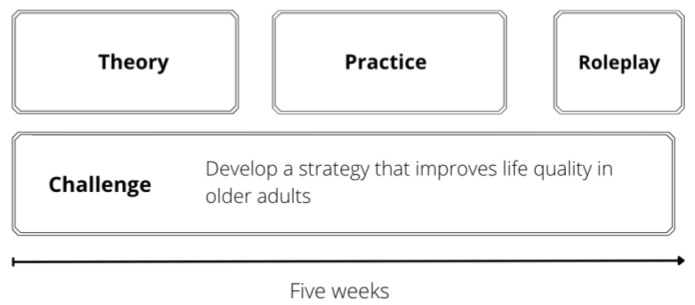
General outline for the Healthy Environments and Self-care for the Older Adults strategy.

**Figure 3 geriatrics-07-00017-f003:**
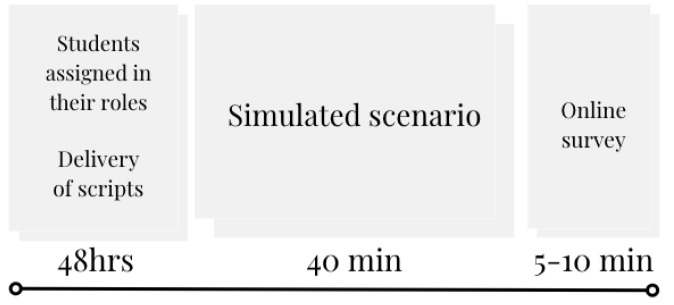
Visual overview of the simulated scenario activity.

**Figure 4 geriatrics-07-00017-f004:**
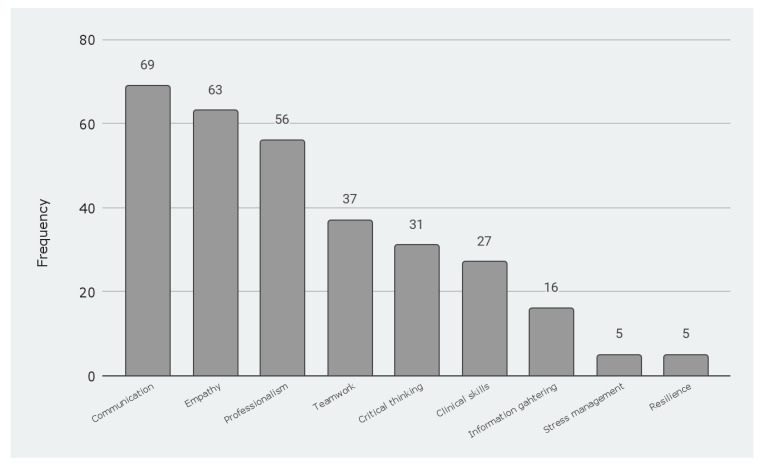
Competencies developed by the educational strategy.

**Table 1 geriatrics-07-00017-t001:** Analytics of the Healthy environments and self-care for the older adult’s strategy.

	Week 1: Physiology of Aging.	Week 2: Comprehensive Care Focused on Older Adults.	Week 3: Caring for the Older Adult	Week 4: Quality of Care	Week 5: Integration
Theoretical Modules or Topics or Activities	1.1 Theories of aging	2.1 Healthy Aging	3.1 Psychological and spiritual care	4.1 General standards of care for the patient.	
	1.2 Physiology of aging by apparatus and systems and its impact on the functionality	2.2 Integrated Care of Older People-ICOPE	3.2 Support networks	4.2 Assessment of the intra- and extra-domiciliary environment.	
	1.3 Types of aging: usual, satisfactory, and pathological	2.3 Medical care	3.3 Psychological and behavioral symptoms in older adults	4.3 Acute and chronic institutionalization.	
	1.4 Quality of life	2.4 Nutritional care			
		2.5 Dental care			
Practical Modules	1.1 Genetic programming	2.1 Comprehensive geriatric assessment	3.1 Depression and anxiety.	4.1 Usefulness of the interdisciplinary approach to improve early diagnosis and treatment of pathologies associated with loss of independence.	5.1 Simulated scenario using role-play dynamic
	1.2 Telomere shortening	2.2 Assessment to Intrinsic Capacity by Integrated Care to Older People	3.2 Dementia.	4.2 Methodology for the design of care guidelines that allow the adequate management of the geriatric patient in different care settings.	
	1.3 Oxidative stress and free radicals	2.3 Identification of heterogeneity in the older adult population (healthy, frail and geriatric patient).	3.3 Caregiver overload		
	1.4 Endocrine theory				
Challenge	1.1 Relationship between the quality of life and health	2.1 Brainstorming of innovative proposals	3.1 Design of the implementation plan (with an established format, recorded in a checklist).	3.2 Integrative project design and implementation plan.	5.1 Presentation of the proposal to internal and external evaluators.
	1.2 Determinants of an adequate quality of life and factors that deteriorate it.	2.2 Selection of the setting in which to develop the project (institution, home, association, community, day center, clinic)		4.1 Report integration.	
	1.3 Quality of life as a goal for healthy living in older people.	2.3 Description of indicators (baseline, intermediate and final) to demonstrate the impact of your project on health			
	1.4 The bio-psycho-social approach to health from the perspective of Intrinsic ability to achieve an adequate quality of life				
	1.5 Situational diagnosis in your community				
	1.6 Development of the theoretical framework				
	1.7 Establish the project’s objective				

**Table 2 geriatrics-07-00017-t002:** Impact of the experience in participant’s training.

Role	Code	Examples of Participants Quotes
Health professional	Clinical needs	*That there are [clinical] needs that affect older adults and can be overlooked by health professionals but they are very important to the patient’s health and dignity (participant health professional #24).*
	Doctor–patient communication	*To integrate, understand, and not ignore what an older adult says (participant health professional #29).*
Observer	Empathy to older adults	*I saw the importance of being empathic to older adults and helping them achieve a healthy aging process. Many things are normalized and attributed to their age without basis (participant observer #17).*
		*It helped me to know how to treat older adults in a professional and empathic manner as well as to attend to their health needs (participant observer #46).*
	Prejudice against older adults	*I observed in detail to avoid prejudices that exist towards older adults (participant observer #12).*
Patient	Empathy to older adults	*I am aware of how older people feel and what they must live with when it comes to non-empathic professionals. This dynamic helped me to be more empathic and to be attentive to details (participant patient #7).*
	Validating emotions	*I got to reflect on how older patients may feel their symptoms or emotions are not taken seriously when attending a consultation. Some professionals or people around them may ignore their symptoms or make them think that it’s because of their age. Now I feel more conscious about the importance of validating and ensuring the adequate health and wellness of the older adult. (Participant patient #15).*
	Prejudice against older adults	*I realized how often health professionals overlook the complaints of their patients just because they come from an older adult. We must be very attentive to what our patient says and take it into account (Patient 18).*

## Data Availability

Not applicable.

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
