# Peer review of "Ageist No More: Interprofessional Training for Undergraduate Healthcare Students"

_geriatrics, 2022, doi:10.3390/geriatrics7010017_

Round 1

Reviewer 1 Report

A brief summary This article contains important information about a clinically and sociological relevant topic that is consistent with the theme of the special issue of the journal. It reports on a course given to undergraduate health science students which was designed to improve their attitudes and competencies in caring for older adult patients.   It  aims to describe  the impact of the course on the students attitudes

The importance of the topic is clear and the description of the course is adequate, but could be improved, particularly given the remote learning design necessitated by the COVID situation. The rationale for adopting the 3 roles needs to be stated explicitly, and the structure of the role assignment needs to be described much more clearly.  Was each student assigned only one role (professional, observer, patient) or could they be assigned more than one role?  Did each “team” consist of one professional, one observer, and one patient?   Were they on Zoom together with the simulated patient and could they all see and hear each other?  Were there any language differences?  How was the background  (lines 108ff) presented to the team (by a narrative from the instructor, or by reading a medical record).  What is the significance of the “observer” in the clinical simulation context?  For example would  the “observer” be considered someone like a family member accompanying the patient, or would this role represent someone assessing the interaction between the clinical professional and patient?  The context and setup for the clinical simulation needs contextualizing.  What was the “Script” that was given them?

The structure of the survey needs to be described more fully

Regarding the sample of 138 students:   It is unclear if these are all the individuals enrolled in or completing the course, or only a subsample of them.  Was there  a nonresponse problem.  The demographic and disciplinary backgrounds of the students would help

Were the nine competencies listed in Figure 2 and exhaustive list of competencies presented in the survey? If so say nine compentecies in line 113.   Were these competencies selected by the curriculum designers based on some pedagogical model or rationale, If so document or describe

In Figure 2 there are a total of 309 responses. If each of the 138 students checked 3 responses, then there should have been 414 responses.  Clearly some respondents may not have indicated any gains in competencies, or only checked one or two. Nonetheless, this suggests that quite a few of the respondents checked fewer than 3 responses.  This should be explained.

Table 2 provides some percentages that do not add up What is being percentaged?  students or comments?.  It is unclear to this reviewer what these percentages represent.  Assuming that each student could only have one role.  It is also unclear what the “codes” in Table 2 represent or how they were derived by the three co-author reviewers.—or how many codes were used.  How the conclusions linking the type of student role played in the exercise  to the kind of change in attitude were arrived at is not well expressed.  Unless this is explained and justified, the conclusions cannot be supported.

The open ended “codes” are not described at all

Inaccuracies/ editing

Line 45- specify ref IF ADDITIONAL CITATION(s) are expected.

Line 144-148  is an incomplete sentence needs to be expanded. .

Author Response

We are pleased with the comments of the expert reviewers that found our manuscript interesting and suitable for publication in the special issue: Ageism, the Black Sheep of the Decade of Healthy Ageing in Geriatrics. Please find below the answers point by point to the concerns raised by the expert reviewers. All concerns are highlighted in the manuscript.

Reviewer 1

A brief summary This article contains important information about a clinically and sociological relevant topic that is consistent with the theme of the special issue of the journal. It reports on a course given to undergraduate health science students which were designed to improve their attitudes and competencies in caring for older adult patients. It aims to describe the impact of the course on the student's attitudes.

Concern 1: The importance of the topic is clear and the description of the course is adequate but could be improved, particularly given the remote learning design necessitated by the COVID situation. The rationale for adopting the 3 roles needs to be stated explicitly, and the structure of the role assignment needs to be described much more clearly. Response: We thank the reviewer for the comments. We have expanded the course description under the subhead “Our Educational Strategy”. We have changed the format of the section with subheadings and substantially expanded the description. You can find it highlighted on page 4. 

  • 1.1 Was each student assigned only one role (professional, observer, patient), or could they be assigned more than one role? We thank the reviewer for the comment, we would like to clarify that the students only had one role during the peer role-play session, we have clarified this in the manuscript by adding sections explaining each role. You can find under the subhead “Description of the role-play” in the Materials and Methods section of the 7th page.
  • 1.2 Did each “team” consist of one professional, one observer, and one patient? Response: We thank the expert reviewer for the comment. Students worked with the same team during the challenge and the role-play. One of them was assigned as the health professional, the other as the patient, and the rest as observers who had a passive role. This has been included in the manuscript in the materials and methods section. Subsections explaining each role have also been included to improve the clarity of the methodology 
  • 1.3 Were they on Zoom together with the simulated patient and could they all see and hear each other? Response:  We thank the expert reviewer for the comment. Students were assigned to Breakout rooms. This has been clarified in the Materials and Methods section.
  • 1.4 Were there any language differences?  Response: We thank the reviewer for the comment and we would like to clarify that there were no language differences between our students and teacher. The experience was conducted in Spanish, the native language of both students and teachers. This has also been clarified in the Materials and Methods section
  • 1.5 How was the background  (lines 108ff) presented to the team (by a narrative from the instructor, or by reading a medical record)?  Response: We thank the reviewer for the comment and would like to clarify the background  The background was given by narrative from the Challenge Advisor to all teams in the pre-briefing time before they start in their breakout rooms. This is included in the Material and Methods section, under the title description of the role-play of the paper.
  • 1.6 What is the significance of the “observer” in the clinical simulation context?  For example, would the “observer” be considered someone like a family member accompanying the patient, or would this role represent someone assessing the interaction between the clinical professional and patient?  The context and setup for the clinical simulation need contextualizing. Response: We thank the reviewer for the comment and we would like to clarify that, in this particular case, the observer had to observe the interaction between the health professional and the patient. He or she had a rubric that focused on key aspects of a clinical consultation like assuring confidence to the patient, voice tone and gestures, including the patient in future plans, and providing a safe place for the patient to express themselves. While this may seem like a passive role that may not obtain benefit from the interaction, our results indicate otherwise. We have included an explanatory paragraph under the subhead c)the observer on the 7th page, Materials, and Methods section.
  • 1.7 What was the “Script” that was given to them?  Response: We agree with the expert reviewer that more information was needed in this section. We have described the script according to the given role in the subheads a, b, and c of the Materials and Methods.

Concern 2: The structure of the survey needs to be described more fully.  Response: We thank the reviewer for the comment. We agree and have explained the survey in the Materials and Methods section, under the subhead “d) the survey”. You may find it highlighted in the manuscript.

Concern 3: Regarding the sample of 138 students: It is unclear if these are all the individuals enrolled in or completing the course, or only a subsample of them.  Was there a nonresponse problem?  The demographic and disciplinary backgrounds of the students would help.  Response: We thank the reviewer for the comment and we have described the background formation of our population in the second paragraph of the 7th page. We did not have a non-response problem

Concern 4: Were the nine competencies listed in Figure 2 and an exhaustive list of competencies presented in the survey? If so say nine competencies in line 113.   Were these competencies selected by the curriculum designers based on some pedagogical model or rationale, if so document or describe.  Response: We thank the reviewer for the comment. We have included de word nine. We have also mentioned and cited the learning frameworks that were used as a basis for selecting these competencies.

Concern 5: In Figure 2 there are a total of 309 responses. If each of the 138 students checked 3 responses, then there should have been 414 responses.  Clearly, some respondents may not have indicated any gains in competencies, or only checked one or two. Nonetheless, this suggests that quite a few of the respondents checked fewer than 3 responses.  This should be explained.  Response: We thank the reviewer and agree. This was one of the limitations, as we programmed the form, we enabled the checkbox option and told participants to select 1-3 responses, and no validation was programmed. We believe that it is important to present the analysis in terms of frequency, then readers can still find a snapshot of what competencies received the most mentions. 

Concern 6: Table 2 provides some percentages that do not add up. What is the percentage?  students or comments?.  It is unclear to this reviewer what these percentages represent.  Assuming that each student could only have one role.  It is also unclear what the “codes” in Table 2 represent how they were derived by the three co-author reviewers.—or how many codes were used.  How the conclusions linking the type of student role played in the exercise to the kind of change in attitude were arrived at is not well expressed.  Unless this is explained and justified, the conclusions cannot be supported.  Response: We thank the reviewer and we agree with the expert reviewer, the format we used in the table was unfitted to represent what we wanted to portray. Therefore we modified and presented the percentages to depict the distribution of roles that participants had. The codes have been explained and the paragraph highlighted before table 2.

Concern 7: The open-ended “codes” are not described at all.  Response: We thank the reviewer, we have included an explanatory paragraph before table 2.

Inaccuracies/ editing We thank the reviewer and we have thoroughly reviewed our manuscript to improve coherence and readability.

Concern 8: Line 45- specify ref IF ADDITIONAL CITATION(s) is expected.  Response: We thank the reviewer for the comment and we would like to clarify that this was an editing mistake, we have deleted the [ref] and searched the manuscript for further mistakes.

Concern 9: Line 144-148  is an incomplete sentence that needs to be expanded.  Response: We thank the reviewer for the comment and have rephrased the paragraph to improve understanding.

We would like to thank Reviewer 1 for the valuable comments, suggestions, and insights on our work. We think our manuscript has considerably improved.

Reviewer 2 Report

  1. In the introduction, line 27 - the authors begin by talking about increasing life expectancy and then suddenly describe how discriminatory attitudes toward older adults "need to be ceased" but there is nothing that yet describes negative attitudes as problematic. Please connect these ideas. Again, in line 30, the "battle" is alluded to before defining the problem that needs to be addressed
  2. Line 34, please provide a description of how not being aware of ageist attitudes represent the worst part. Provide examples of the consequences. 
  3. Line 34, there is no connection between not being aware of attitudes to healthcare workers being at risk. Please connect these concepts more meaningfully.
  4. Line 42 - instead of "with" it  is more appropriate to phrase it as negative attitude toward or against older adults
  5. 1.1 line 68 - replace "scales of ageism" with rates of ageism
  6. Line 83 - awkward sentence please consider revising- First, they must investigate the quality of life in aging to older family or community members by applying the quality-of-life assessment questionnaire to older adults
  7. Methods - page 95 - This study has both quantitative and qualitative components as the authors report out on survey data so it is no solely a qual study as indicated.
  8. line 147 - the sentence is incomplete and seems to be missing a portion.
  9. Given the primary findings that the top three competencies were communication, empathy and professionalism, more attention should be given to these constructs in the introduction and discussion sections. While they are alluded to wth references in the discussion section, it would be appropriate to first mention these and highlight their importance in the introduction section. 
  10. The authors do not mention any limitations to the study or the results

Author Response

We are pleased with the comments of the expert reviewers that found our manuscript interesting and suitable for publication in the special issue: Ageism, the Black Sheep of the Decade of Healthy Ageing in Geriatrics. Please find below the answers point by point to the concerns raised by the expert reviewers. All concerns are highlighted in the manuscript.

Concern 1: In the introduction, line 27 - the authors begin by talking about increasing life expectancy and then suddenly describe how discriminatory attitudes toward older adults "need to be ceased" but there is nothing that yet describes negative attitudes as problematic. Please connect these ideas. Again, in line 30, the "battle" is alluded to before defining the problem that needs to be addressed.  Response: We would like to thank the reviewer for the comment. We have included a paragraph defining negative attitudes towards older adults and how they translate and harm the quality of life of older adults. The second page, first paragraph.

Concern 2: Line 34, please describe how not being aware of ageist attitudes represents the worst part. Provide examples of the consequences.  Response: We thank the expert reviewer for the comment, we have expanded this section and included the description of stereotypes and how they contribute to detrimental attitudes towards older adults. You can find it highlighted in the XX section.

Concern 3: Line 34, there is no connection between not being aware of attitudes to healthcare workers being at risk. Please connect these concepts more meaningfully.  Response: We thank the reviewer for the comment. We have explained how ageist attitudes put patients at risk and expanded the whole paragraph. You can find it highlighted in the second paragraph of the second page.

Concern 4: Line 42 - instead of "with" it is more appropriate to phrase it as a negative attitude toward or against older adults.  Response: We thank the reviewer for the comment. We have rearranged the whole paragraph to focus on the main idea about the importance of an early introduction to offer care to the elder to counteract ageism.

Concern 5: 1.1 line 68 - replace "scales of ageism" with rates of ageism.  Response: We thank the reviewer for the comment. The whole paragraph was changed to improve coherence and readability.

Concern 6: Line 83 - awkward sentence please consider revising- First, they must investigate the quality of life in aging to older family or community members by applying the quality-of-life assessment questionnaire to older adults.  Response: We thank the reviewer for the comment, we have clarified that the older adults belong to a marginalized community which beforehand we know are deprived of a decent quality of life, highlighted, XX section.

Concern 7: Methods - page 95 - This study has both quantitative and qualitative components as the authors report out on survey data so it is not solely a qual study as indicated.  Response: We thank the reviewer for the comment, and we have pointed out at the end of methods that after categorizing the results, quantitative analysis was also done.

Concern 8: line 147 - the sentence is incomplete and seems to be missing a portion.  Response: We thank the reviewer for the comment, we agree and have rephrased the complete paragraph to make sense.

Concern 9: Given the primary findings that the top three competencies were communication, empathy, and professionalism, more attention should be given to these constructs in the introduction and discussion sections. While they are alluded to with references in the discussion section, it would be appropriate to first mention these and highlight their importance in the introduction section.  Response: We thank the reviewer for the comment, and we have added a paragraph discussing how communication, empathy, and professionalism are key abilities in patient-centered clinical care and how this helps to construct a healthy relationship between health professionals and patients, including references. You can find it highlighted in the Materials and Methods section, under the subhead “d) the survey”.

Concern 10: The authors do not mention any limitations to the study or the results.  Response: We thank the reviewer for his/her comments, we have included a paragraph and extensively discussed the limitations of the study. You can find it highlighted before the Conclusions.

We would like to thank Reviewer 2 for the valuable comments, suggestions, and insights on our work. We think our manuscript has considerably improved.

Reviewer 3 Report

Thank you for giving me the opportunity to reviewthsimanuscript. However, all sections of the manuscript are very brief and not sufficient. The significance of the study is questionable. The methods used can be clearer.

Author Response

We are pleased with the comments of the expert reviewers that found our manuscript interesting and suitable for publication in the special issue: Ageism, the Black Sheep of the Decade of Healthy Ageing in Geriatrics. Please find below the answers point by point to the concerns raised by the expert reviewers. All concerns are highlighted in the manuscript.

Reviewer 3

Concern 1: Thank you for giving me the opportunity to review the manuscript. However, all sections of the manuscript are very brief and not sufficient. The significance of the study is questionable. The methods used can be clearer. Response: We thank the expert reviewer for his/her comments. We have thoroughly revised our manuscript, enriching, and expanding all the sections included and designing figures to improve the quality of our work. This manuscript represents three years of work to build an innovative educational strategy to prepare undergraduate healthcare students in terms of aging knowledge by biological, psychological, and social features focusing on the quality of life and always with a social impact to achieve health professionals with the tools to care for the older persons without stigma, prejudice, and discrimination, thereby reducing stereotype threat to older persons, edadism and then improve the patient-centered care.

We have discussed the importance of our work in the manuscript in the conclusions section.

We would like to thank Reviewer 3 for the valuable comments, suggestions, and insights on our work. We think our manuscript has considerably improved.

Round 2

Reviewer 1 Report

Vastly improved.  Perhaps some of the evaluative remarks in the presentation of findings might better be moved to the final section.

Reviewer 3 Report

The authors have put great effort into the revised manuscript. Good work!